# Comparative study on the changes of bacterial species and severity of antimicrobial resistance during 13 years

**Huili Zhang** [1]*, **Kairui Zhou** [1], **Xinglong He** [2], **Xin Yuan** [1]

**1** Department of Pharmacy, The Second People's Hospital of Xinxiang, Xinxiang, Henan, China, **2** Clinical Laboratory, The Second People's Hospital of Xinxiang, Xinxiang, Henan, China

☯ These authors contributed equally to this work.
* zhlyzy218@163.com

## Abstract

### Background

With the widespread use of broad-spectrum antibiotics, the problem of bacterial resistance has become a global crisis. To monitor bacterial resistance in our hospital, the distribution of specimens, the detection of pathogens and their drug resistance from July 2005 to June 2007 (13 years ago) and July 2018 to June 2020 were compared and analyzed.

### Methods

Ordinary specimens (such as sputum, urine, feces, and secretion) were inoculated in blood AGAR media, MacConkey medium, chocolate medium, double SS medium and selective culture medium. Blood, cerebrospinal fluid, pleural effusion, joint cavity effusion and other sterile body fluid samples were inoculated in aerobic and anaerobic blood culture flasks. Automatic microbial identification, drug sensitivity analysis and mass spectrometry analysis were used to determine their drug sensitivity.

### Results

Compared with the results obtained 13 years ago, the number of specimens submitted for inspection in the past two years has increased significantly, exhibiting a growth rate of 283%. The changes in the pathogen species were obvious. Gram-positive cocci were the dominant bacteria 13 years ago, and Gram-negative bacilli were the dominant bacteria in the past two years. In addition, the resistance of several major Gram-negative bacilli to piperacillin/tazobactam, cefoperazone/sulbactam, meropenem and imipenem all showed an increasing trend.

### Conclusion

The variety of pathogenic bacteria in our hospital has changed significantly in the past two years compared with that 13 years ago, and the clinical isolates of Gram-negative bacilli have increased significantly compared with Gram-positive cocci. In the clinical treatment of

**Data Availability Statement:** All relevant data are within the manuscript.

**Funding:** The author(s) received no specific funding for this work.

**Competing interests:** The authors have declared that no competing interests exist.

anti-infective diseases, antimicrobial agents should be selected according to the bacterial distribution characteristics and drug resistance in each hospital.

## Introduction

The incidence of microbial infections has been increasing in the past several decades, and antibacterial drugs are widely used clinically as the first-line therapy to inhibit the growth and reproduction of bacteria; this, in turn, has led to the emergence of specific drugs and multidrug resistance among various strains of microorganisms [1]. The problem of bacterial resistance has become a global crisis; the World Health Organization (WHO) published the first global antibiotic resistance report based on data from 114 countries around the world in 2014 [2]. In 2020, the WHO released an activity with the theme of "Unite to Protect Antimicrobial Drugs" to raise the public and medical staff's awareness of the drug resistance crisis through extensive publicity on the rational use of antimicrobial drugs. Resistance has important implications for clinicians and patients due to a higher risk of inadequate treatment, an increased length of hospital stay, and additional healthcare costs [3].

Among clinical isolates, the ratio of Gram-negative bacilli to Gram-positive cocci is approximately 7 to 3 according to the CHINET surveillance from 2005 to 2017 and the CARSS surveillance in 2016 [4, 5]. Multidrug-resistant Gram-negative bacterial infections are considered one of the major threats to global health. They are the leading causes of nosocomial infections around the world [6–8]. This makes it particularly important to monitor the distribution of clinical isolates and the changes in drug resistance.

Among Gram-positive cocci and Gram-negative bacilli with high clinical isolation rate, *Staphylococcus* aureus mainly produces three kinds of toxins (Pore-forming toxins, Exfoliative toxins, Superantigens) to degrade host cells and weaken the body's response, thus causing skin and soft tissue infections and lung infections [9]. The virulence factors (toxins or biofilm) of Gram-negative bacteria such as *Escherichia coli*, *Acinetobacter baumannii*, and *K. pneumonia* can participate in bacterial adhesion, invasion, and escape from the host's immune defense, causing the destruction of host cells or tissues, thus cause lung infection, bloodstream infection, abdominal infection and urinary tract infection [10, 11]. These bacteria can lead to multidrug resistance through hydrolase production, efflux pump overexpression or membrane pore protein mutation [12, 13]. Multidrug resistance has been increased globally that is considered a public health threat. Several previous studies revealed the emergence of multidrug-resistant bacterial pathogens from different origins especially fish, birds, animals, and food chains which may be transmitted to human consumers resulting in severe illness [14–16].

Here, the source and species of pathogen specimens, the number of detected pathogens and the change of drug resistance in our hospital from July 2018 to June 2020 and from July 2005 to June 2007 were statistically analyzed. It provides a theoretical basis for clinicians to choose antibacterial drugs, so as to reduce the unreasonable use of antibacterial drugs and effectively control the bacterial drug resistance in our hospital.

## Materials and methods

### Source of specimens

The research specimens were from the sterile body fluid samples (blood, cerebrospinal fluid, pleural effusion, joint cavity effusion, etc.) and common specimens (sputum, urine, feces,

secretions, etc.) which sent by all clinical departments of our hospital to the microbiology room for bacterial culture from July 2018 to June 2020 and from July 2005 to June 2007. The examination was submitted in accordance with the normal procedures of clinical examination, and the collection and use of relevant test results and data were approved by the Medical Ethics Committee of our hospital.

## Source of bacteria

The pathogenic bacteria detection results from our hospital's microbiology laboratory from July 2018 to June 2020 were selected to perform statistics and to compare and analyze the data from July 2005 to June 2007 [17].

## Source of medium and reagent plate

The bacterial culture plate containing different media was purchased from Antu Bio-Engineering Co., LTD, Zhengzhou, China, and the random in vitro diagnostic reagent plate of the bacterial determination system was purchased from Deere Bio-Engineering Co., LTD, Zhuhai, China.

## Isolation and identification of bacterial pathogens

Sterile body fluid specimens, such as blood, cerebrospinal fluid, pleural and ascites fluid, and joint cavity effusions, were inoculated into aerobic and anaerobic blood culture bottles and placed in a BD9120 automatic blood incubator. The blood bottle specimens were considered positive according to the normal specimen process. Various common specimens were inoculated on blood agar medium, MacConkey medium, chocolate medium, double SS medium, and other basic and selective media in a 35~37°C, 5~10% $CO_2$ incubator for 18~24 h. Separately, according to the routine methods, Gram stain smears, oxidase tests, and thiolase tests were applied, and then we used a DL-96II automatic microorganism identification and drug sensitivity analyzer(the specimens from 2005 to 2007 used the Black Horse Bact-IST automatic microbial analysis system) and a VITEK MS mass spectrometer and proceeded according to the operating instructions. The bacteria were identified according to the biochemical reaction provided by the random in vitro diagnostic reagent plate of the bacterial determination system, and the semi-quantitative analysis of the MIC of antibacterial agents contained in the sensitive cards was performed. The results were analyzed according to the Clinical and Laboratory Standards Institute (CLSI) 2018 definitions [18].

## Statistical analysis

Statistics on the source and number of pathogenic bacteria specimens and the type, number, and drug resistance of the pathogenic bacteria detected in the pathogenic microorganism laboratory from July 2018 to June 2020, a comprehensive analysis of the characteristics of the pathogenic bacteria detected in our hospital and the status of their antimicrobial resistance, and a comparative analysis with the data from July 2005 to June 2007 were performed to learn about the sources of pathogenic bacteria specimens, the changes in species over time, the number of pathogens detected and the changes in drug resistance over time in order to provide a reference for the rational application of antimicrobial drugs in the hospital. Office 2016 software was used to perform statistical analysis and to draw charts about the sources of the specimens, types, and numbers of pathogens.

Table 1. The distribution of specimen types before and after 13 years.

| Specimen source | Number of bacteria 13 years ago | Percentage % | Number of bacteria at present | Percentage % | Rate of increase % |
|---|---|---|---|---|---|
| Sputum | 1296 | 37.3 | 5450 | 41.3 | 320.5 |
| Blood | 956 | 27.8 | 4374 | 33.1 | 357.5 |
| Urine | 240 | 7.0 | 1066 | 8.1 | 344.2 |
| Secretions | 948 | 27.5 | 1676 | 12.7 | 76.8 |
| Feces | -- | -- | 153 | 1.2 | -- |
| Others | -- | -- | 483 | 3.7 | -- |
| Total | 3440 | 100 | 13202 | 100 | 283.8 |

## Results

### Distribution of the specimen types to be examined

Our hospital sent a total of 13,202 pathogenic bacteria cultures from July 2018 to June 2020, an increase of 283.8% compared with 13 years ago. The samples were still mainly sputum 41.3% (5,450/13202), blood 33.1% (4374/13202), secretions 12.7% (1676/13202) and urine 8.1% (883/13202) (Table 1), similar to the distribution of specimen types thirteen years ago.

### Species and quantity of pathogenic bacteria

From July 2018 to June 2020, 2956 strains of pathogenic bacteria were cultured in our hospital, and 1616 strains were cultured 13 years ago, an increase of 82.92% over the previous period. The types and number of pathogenic bacteria detected are shown in Table 2. As shown in the table, thirteen years ago, the pathogen was identified in the majority of the Gram-positive cocci. The overall detection rate was 39.10% (*Staphylococcus aureus* 31.40%), and the Gram-negative bacilli overall detection rate was 32.50% (*Escherichia coli* 12.90%, *Pseudomonas aeruginosa* 11.40%, *Klebsiella pneumoniae* 8.20%). The detection rate of the *fungi* was also high, accounting for 18.20% of the total detected pathogens. From September 2018 to June 2020, Gram-negative bacilli were dominant, with a total detection rate of 62.86%, among which *K. Pneumoniae* was 22.83%, *P. Aeruginosa* was 17.66%, and *E. Coli* was 13.44%. The total detection rate of Gram-positive cocci was 19.35%, of which the detection rates of *S. Aureus* and *Coagulase-negative staphylococcus (CoNS)* were significantly lower than those 13 years ago, accounting for 9.91% and 3.38% of the total detected pathogens, respectively. The detection rate of fungi was also significantly lower than before.

Table 2. The comparison of types and quantities of pathogenic bacteria before and after 13 years.

| Species | Thirteen years ago | Percentage % | At the current time | Percentage % | The rate of change% |
|---|---|---|---|---|---|
| *Staphylococcus aureus* | 508 | 31.4 | 293 | 9.91 | -42.32 |
| *Coagulase negative staphylococcus* | 124 | 7.7 | 100 | 3.38 | -19.35 |
| *Streptococcus* | -- | -- | 51 | 1.73 | -- |
| *Enterococcus* | -- | -- | 128 | 4.33 | -- |
| *Pseudomonas aeruginosa* | 184 | 11.4 | 522 | 17.66 | 183.70 |
| *Klebsiella pneumoniae* | 132 | 8.2 | 675 | 22.83 | 411.36 |
| *Escherichia coli* | 208 | 12.9 | 397 | 13.44 | 90.86 |
| *Acinetobacter baumannii* | -- | -- | 264 | 8.93 | -- |
| *Fungus* | 304 | 18.8 | 134 | 4.53 | -55.92 |
| Other bacteria | 156 | 9.6 | 392 | 13.26 | 151.28 |
| Total | 1616 | 100 | 2956 | 100 | 82.92 |

**Table 3. The change in the resistance rate (%) of the main Gram-negative bacilli to common antibacterial drugs before and after 13 years.**

| Antibacterial drugs | *Pseudomonas aeruginosa* | | *Klebsiella pneumoniae* | | *Escherichia coli* | | *Acinetobacter baumannii* | |
|---|---|---|---|---|---|---|---|---|
| | before | after | before | after | before | after | before | after |
| Ceftazidime | 16.9 | 48.76 | 59.5 | 45.47 | 30.2 | 33.99 | -- | 77.02 |
| Aztreonam | 32.5 | 42.56 | 65.8 | -- | 33.3 | -- | -- | -- |
| Cefoperazone | 62.3 | -- | 76 | -- | 70 | -- | -- | -- |
| Cefoperazone/Sulbactam | 8.2 | 45.64 | 6.3 | 36.21 | 3.5 | 12.53 | -- | 57.76 |
| Cefotaxime | 40.3 | -- | 65.8 | -- | 77.8 | -- | -- | -- |
| Cefepime | 19.5 | 42.98 | 25.2 | 30.86 | 36.5 | 32.34 | -- | 68.32 |
| Imipenem | 15.6 | 54.96 | 0 | 17.49 | 0 | 3.3 | -- | 70.81 |
| Meropenem | 18.2 | 42.98 | 0 | 21.4 | 0 | 2.97 | -- | 72.67 |
| Piperacillin | 54.5 | 53.72 | 69.4 | -- | 71.4 | -- | -- | -- |
| Piperacillin/tazobactam | 7.8 | 45.04 | 9.9 | 37.65 | 6.3 | 10.56 | -- | 75.61 |
| Amoxicillin Clavulanate potassium | -- | -- | 80.2 | -- | 79.4 | -- | -- | -- |
| Ampicillin | -- | -- | 90.1 | -- | 88.9 | 87.13 | -- | -- |
| Ampicillin/Sulbactam | -- | -- | 79.3 | 55.97 | 84.1 | 39.6 | -- | 78.26 |
| Ceftriaxone | -- | -- | 74.5 | 53.5 | 85.7 | 59.74 | -- | 81.37 |
| Cefuroxime | -- | -- | 82.9 | 56.58 | 87.3 | 63.04 | -- | -- |
| Cefazolin | -- | -- | 87.4 | 58.64 | 87.3 | 65.35 | -- | -- |
| Polymyxin B | -- | 3.31 | -- | -- | -- | -- | -- | 4.35 |
| Amikacin | 36.4 | 23.55 | 65.8 | 26.75 | 30.2 | 8.25 | -- | 73.91 |
| Gentamicin | 81.8 | 42.15 | 80.2 | 44.44 | 68.3 | 45.87 | -- | 80.75 |
| Tobramycin | 83.1 | 49.17 | -- | -- | -- | -- | -- | -- |
| Netilmicin | 77.9 | -- | 70 | -- | 55 | -- | -- | -- |
| Ciprofloxacin | 66.2 | 44.21 | 51.4 | 47.12 | 81 | 65.68 | -- | 78.26 |
| Norfloxacin | 67.5 | 43.15 | 55 | -- | 85.7 | -- | -- | -- |
| Levofloxacin | 49.4 | 45.45 | 53.9 | 42.39 | 80.4 | 62.71 | -- | 71.43 |
| Trimethoprim/sulfamethoxazole | 100 | -- | 57.5 | 53.29 | 82.5 | 69.31 | -- | 73.91 |
| Chloramphenicol | 66.2 | -- | -- | -- | -- | 28.71 | -- | -- |
| Macrodantin | -- | -- | 45.9 | 23.25 | 22.2 | 4.62 | -- | -- |

## Analysis of bacterial resistance

The drug resistance rates of the main Gram-negative bacilli and Gram-positive cocci detected in the microbiology laboratory of our hospital against common antibacterial drugs are shown in Tables 3 and 4. Resistance analysis was performed on the first four Gram-negative bacilli (*K. Pneumoniae*, *P. Aeruginosa*, *E. Coli*, *Acinetobacter baumannii*) and two Gram-positive cocci (*S. Aureus* and *CoNS*).

**Klebsiella pneumoniae.** Compared with 13 years ago, the resistance rate of *K. Pneumoniae* to piperacillin/tazobactam, cefoperazone/sulbactam, cefepime, meropenem, and imipenem has increased significantly. Among them, the resistance rates to meropenem and imipenem have increased from zero 13 years ago to 21.4% and 17.94% now; The resistance rate to cefoperazone/sulbactam, piperacillin/tazobactam and cefepime increased from the previous 6.3%, 9.9%, 25.2% to 36.21%, 37.65%, and 30.86%, respectively, and the rapid increase of resistance indicates that the resistance of *K. Pneumoniae* to antibiotics should not be ignored. The resistance rate to aminoglycosides (gentamicin, amikacin) and the first and second generation cephalosporins has declined.

**Pseudomonas aeruginosa.** Compared with 13 years ago, the drug resistance rates of *P. Aeruginosa* to most antibacterial drugs were significantly increased, among which the drug

**Table 4.  The comparison of changes in the resistance rate (%) of the main G+ cocci to commonly used antibacterial drugs approximately 13 years ago.**

| Antibacterial drugs | *Staphylococcus aureus* | | *Coagulase negative staphylococcus* | |
|---|---|---|---|---|
| | **before** | **after** | **before** | **after** |
| Azithromycin | 45 | 74.16 | 80 | 78.95 |
| Cefoxitin | -- | 25.85 | -- | 6.6 |
| Chloramphenicol | -- | 11 | -- | 18.42 |
| Clindamycin | 37.5 | 72.25 | -- | 60.23 |
| Clarithromycin | 45 | 75.12 | 80 | 77.63 |
| Erythrocin | 50 | 76.08 | 80 | 78.95 |
| Macrodantin | 75 | 2.87 | 80 | 13.16 |
| Gentamicin | 50 | 30.62 | 30 | 38.16 |
| Levofloxacin | 37.5 | 32.54 | 60 | 59.21 |
| Linezolid | -- | 0 | 60 | 0 |
| Moxifloxacin | -- | 30.62 | -- | 48.68 |
| Norfloxacin | 62 | 37.32 | -- | 43.16 |
| Oxacillin | 0 | 25.84 | 70 | 35.53 |
| Penicillin | 87.5 | 90.91 | 0 | 86.84 |
| Rifampicin | -- | 13.4 | 97.9 | 15.79 |
| Trimethoprim/sulfamethoxazole | 75.5 | 40.19 | -- | 55.26 |
| Teicoplanin | -- | 0 | 80 | 0 |
| Tetracycline | -- | 23.44 | -- | 35.53 |
| Tigecycline | -- | 0 | -- | 0 |
| Vancomycin | 0 | 0 | -- | 0 |
| Amikacin | -- | 7.18 | -- | 87.5 |

resistance rates to third- and fourth-generation cephalosporins, carbapenems, cefoperazone/sulbactam, and piperacillin/tazobactam were significantly higher than before, with drug resistance rates of > 40%. The drug resistance rate of aminoglycosides and quinolones decreased. The sensitivity to polymyxin B was still good.

**Escherichia coli.**   Compared with 13 years ago, the drug resistance rates of *E. Coli* to meropenem and imipenem increased from zero resistance to 2.97% and 3.3%, respectively, maintaining a high sensitivity. It also maintains good sensitivity to cefoperazone/sulbactam and piperacillin/tazobactam. The drug resistance rate to first-, second-, and third-generation cephalosporins is still high. Resistance to aminoglycosides and quinolones also remains high.

**Acinetobacter *Baumannii*.**   The drug resistance rate of *A. Baumannii* was not evaluated thirteen years ago. The analysis results of the last two years showed that the drug resistance rate of *A. Baumannii* to most antibacterial drugs was > 50%, and the drug resistance rate of carbapenem was > 70%, showing good sensitivity to polymyxin B.

**Staphylococcus *Aureus*.**   Compared with 13 years ago, the resistance rate of *S. Aureus* to oxacillin increased from zero to 25.84%, the resistance rate to penicillin was more than 90%, and that to clindamycin, clarithromycin, erythromycin, azithromycin also increased, but no *S. Aureus* resistance to vancomycin, teicoplanin, linezolid, or tigecycline was found.

**Coagulase-negative staphylococcus.**   Compared with 13 years ago, the drug resistance rate of *CoNS* to oxacillin increased from zero to 35.53%, and the drug resistance to penicillin, clarithromycin, erythromycin, and azithromycin was still high, with a drug resistance rate of > 75%. No *CoNS* resistant to vancomycin, teicoplanin, linezolid or tigecycline were found.

## Discussion

Comparative analysis of the distribution of the types of specimens submitted for inspection found that our hospital prepared a total of 13,202 pathogenic bacteria cultures from July 2018 to June 2020, an increase of 283% from 13 years ago, indicating that the awareness and number of doctors in the hospital had been significantly increased. The specimens were mainly respiratory tract, blood, secretions and urine, which were similar to those 13 years ago.

Changes of pathogen species were detected. From July 2018 to June 2020, 2956 strains of pathogenic bacteria were cultured in our hospital, an increase of 82.92% compared with 13 years ago. The types of pathogenic bacteria changed significantly. Thirteen years ago, Gram-positive cocci were the majority, with a total detection rate of 39.10%, and the total detection rate of Gram-negative bacilli was 32.50%. In the past two years, Gram-negative bacilli were dominant, with a total detection rate of 62.86%, and the total detection rate of Gram-positive cocci was 19.35%. Similarly, the ratio of Gram-negative bacilli to Gram-positive cocci was approximately 7 to 3 according to CHINET surveillance from 2005 to 2017 and that obtained through CARSS surveillance in 2016 [4, 5]. Based on this result and previous experience, clinicians can determine the common pathogens of different diseases and choose effective antimicrobial treatment.

For nearly two years, the top five pathogens identified in the hospital were *K. Pneumoniae*, *P. Aeruginosa*, *E. coli*, *S. Aureus*, and *A. Baumannii*, and one of the five CHINET statistics in the first half of 2020 (*E. coli*, *K. Pneumoniae*, *S. Aureus*, *P. Aeruginosa*, and *A. Baumannii*) were the same but in a different order. In addition, according to CHINET statistics, since 2017, the isolation rate of *K. Pneumoniae* in respiratory specimens has exceeded that of *A. Baumannii*, rising to first place [19, 20].

An analysis of pathogen resistance to common antibacterial drugs detected in our hospital was conducted. Antibiotic-resistant nosocomial infections pose a serious clinical challenge to doctors in the ICU and other departments, increasing morbidity, mortality, length of stay, and health care costs [21, 22]. At present, antibiotic resistance in our hospital is quite serious, especially in the ICU. The detection rate of *S. Aureus* and *CoNS* resistant to oxacillin was significantly higher than that thirteen years ago, and the detection rate of carbapenem-resistant Gram-negative bacilli was significantly increased. Care should be taken to consider multidrug-resistant bacteria when treating infections in patients in the ICU. Among the Gram-negative bacilli, the drug resistance rate of *P. Aeruginosa* to most antibacterial drugs was significantly increased, the drug resistance rate to third- and fourth-generation cephalosporins, carbapenems, cefoperazone/sulbactam, and piperacillin/tazobactam was significantly increased compared with that before, with a drug resistance rate of > 40%. The WHO reported in 2017 that carbapenem-resistant *P. Aeruginosa* was listed in the "critical" group for which new antibiotics were urgently required [23]. It is suggested that clinicians should avoid using the third and fourth generation cephalosporins and their compound preparations when treating *P. Aeruginosa* infection. The detection rates of carbapenem-resistant *K. Pneumoniae* and *A. Baumannii* were also significantly increased. The drug resistance rates of *K. Pneumoniae* to meropenem and imipenem increased from zero thirteen years ago to 21.4% and 17.94%, respectively, while the drug resistance rate of *A. Baumannii* to carbapenems was > 70%. Colistin maintained very high in vitro antimicrobial activity against *P. Aeruginosa* and *A. baumannii* (more than 95% of isolates exhibited susceptibility at all timepoints). *K. Pneumoniae* and *P. Aeruginosa* were highly resistant to aminoglycosides thirteen years ago in our hospital. In recent years, due to their toxicity and side effects, aminoglycosides have mainly been used to jointly control infections of multidrug resistant bacteria [24]. Therefore, the antimicrobial resistance of *K. Pneumoniae* and *P. Aeruginosa* to aminoglycosides has decreased in our hospital in the last two

years. *E. Coli* was highly sensitive to meropenem and imipenem, with a sensitivity rate of > 96%. Although the detection rate of Gram-positive cocci in our hospital was significantly lower than that 13 years ago, its resistance to oxacillin increased significantly, and the resistance rate of *S. Aureus* to oxacillin increased from zero to 25.84%; penicillin, clarithromycin, erythromycin, and azithromycin resistance remained high. No resistance was found against vancomycin, teicoplanin, and linezolid. Vancomycin has historically been the drug of choice for the treatment of methicillin-resistant *S. Aureus* infections, but its increased use has already led to vancomycin-intermediate *S. aureus* (VISA) as well as vancomycin-resistant *S. aureus* (VRSA) in certain parts of the world [25–27].

Challenges posed by multidrug resistant and pandrug resistant bacteria have become evident in recent years with the proliferation of various multidrug-resistant Gram-negative bacteria, such as extended-spectrum beta-lactamase (ESBL)-producing Enterobacterales, carbapenem-resistant Enterobacterales (CRE), carbapenem-resistant *Acinetobacter baumannii* (CRAB), carbapenem-resistant *P. Aeruginosa*, and other carbapenem-resistant Gram-negative bacteria, has introduced new challenges to clinical anti-infectious disease treatment and hospital infection control [28–32]. According to the statistics, approximately 50% of *K. pneumoniae* strains produce ESBLs, and a marked change is that carbapenem-resistant *K. pneumoniae* (CRKP) increased from 3.0% in 2005 to 20.9% in 2017 [4]. In China, the dominant genotype of CRKP is *K. Pneumoniae* carbapenemase-2 (KPC-2), accounting for approximately 70% of cases [33]. Because carbapenem-resistant Gram-negative bacilli are usually extensively drug-resistant, infections caused by these drug-resistant bacteria are difficult to treat, including challenges related to diagnosis and treatment, and they cause increased morbidity and mortality [34, 35].

The monitoring of bacterial resistance in medical institutions, regions, and nationwide is helpful to grasp the sensitivity of clinically important pathogenic bacteria to antibacterial drugs and to provide a basis for the empirical treatment of infections. Our data show significant changes and trends in the drug resistance of clinically important pathogenic bacteria, which provides a reference for hospitals to comprehensively understand drug resistance. To promote the rational use of antimicrobial drugs and to reduce the occurrence of drug resistance, personnel at all levels should be trained in the clinical application and management of antimicrobial drugs. In addition, while selecting antibacterial drugs based on the results of drug sensitivity, it is also recommended to combine pharmacokinetic/pharmacodynamic (PK/PD) data and MIC determination to calculate the correct dose of antibacterial drugs for each patient [36].

## Conclusion

The variety of pathogenic bacteria in our hospital has changed significantly in the past two years compared with that 13 years ago, and the clinical isolates of Gram-negative bacilli have increased significantly compared with Gram-positive cocci. In the clinical treatment of anti-infective diseases, antimicrobial agents should be selected according to the bacterial distribution characteristics and drug resistance in each hospital. In addition, bacterial resistance monitoring at the hospital level is an important part of antimicrobial management measures in China, and local antimicrobial resistance data are crucial to guide the rational use of antimicrobial agents.

## Acknowledgments

We thank all those involved in this study.

## Author Contributions

**Conceptualization:** Huili Zhang.

**Data curation:** Huili Zhang, Kairui Zhou.

**Investigation:** Huili Zhang, Kairui Zhou.

**Methodology:** Xinglong He.

**Project administration:** Huili Zhang.

**Resources:** Huili Zhang, Xinglong He.

**Supervision:** Huili Zhang, Xin Yuan.

**Validation:** Huili Zhang, Kairui Zhou.

**Visualization:** Huili Zhang, Kairui Zhou.

**Writing – original draft:** Kairui Zhou, Xin Yuan.

**Writing – review & editing:** Huili Zhang, Kairui Zhou.

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
