## [Decision Letter · Decision Letter 0]

23 Jun 2021

PONE-D-21-10409

Comparative study regarding the differences in bacterial resistance after approximately 13 years

PLOS ONE

Dear Dr. Zhang,

Thank you for submitting your manuscript to PLOS ONE. After careful consideration, we feel that it has merit but does not fully meet PLOS ONE’s publication criteria as it currently stands. Therefore, we invite you to submit a revised version of the manuscript that addresses the points raised during the review process.

ACADEMIC EDITOR: A major revision is needed.

We look forward to receiving your revised manuscript.

Kind regards,

Abdelazeem Mohamed Algammal, Prof, Ph.D

Academic Editor

PLOS ONE

Journal Requirements:

Reviewers' comments:

Reviewer's Responses to Questions

**Comments to the Author**

1. Is the manuscript technically sound, and do the data support the conclusions?

Reviewer #1: Yes

2. Has the statistical analysis been performed appropriately and rigorously? 

Reviewer #1: Yes

3. Have the authors made all data underlying the findings in their manuscript fully available?

Reviewer #1: Yes

4. Is the manuscript presented in an intelligible fashion and written in standard English?

Reviewer #1: No

5. Review Comments to the Author

Reviewer #1: Comments to authors:

- The current study is interesting; however, the authors should address the below-outlined comments:

- The manuscript should be revised for language editing and grammar mistakes.

-Please write the scientific names of bacterial pathogens in correct form all over the manuscript (italic form).

Title:

I think the work would benefit from the title that contains main conclusion of the study (should be derived from the conclusion), please modify the title.

Abstract:

- The abstract must illustrate the main conclusion of your study (please improve).

Introduction: (It needs to be more informative)

-Please give a hint about the virulence determinants, the pathogenesis, and diseases caused by S. aureus, E. coli, Acinetobacter baumannii, and K. pneumonia.

-The authors should illustrate the public health importance concerning the emergence of multidrug-resistant (MDR) bacterial pathogens. Several studies proved the widespread MDR- bacterial pathogens;

Authors could add the following paragraph and use the following references:

Multidrug resistance has been increased globally that is considered a public health threat. Several previous studies revealed the emergence of multidrug-resistant bacterial pathogens from different origins especially fish, birds, animals, and food chains which may be transmitted to human consumers resulting in severe illness.

You could use and cite the following valuable recent studies:

1-PMID: 32497922 ; https://pubmed.ncbi.nlm.nih.gov/32497922/

2-PMID: 32397408 ; https://www.mdpi.com/2076-0817/9/5/362

3-PMID: 30150182 ; https://pubmed.ncbi.nlm.nih.gov/30150182/

4-PMID: 32532070 DOI: 10.3390/toxins12060383

5-PMID: 32235800 DOI: 10.3390/pathogens9030238

6-PMID: 32472209 DOI: 10.1186/s13568-020-01037-z

7-PMID: 32994450 DOI: 10.1038/s41598-020-72264-4

8-El-Sayed M, Algammal A, Abouel-Atta M, Mabrok M, Emam A. Pathogenicity, genetic typing, and antibiotic sensitivity of Vibrio alginolyticus isolated from Oreochromis niloticus and Tilapia zillii. Rev. Med. Vet. 2019 Jan 1; 170:80-6.

-Rephrase the aim of work to be clearer and sound better.

Material and methods

-Add more data about the Source of specimens (sex, age, disease…ect.)

- The subtitle: Bacteria isolation and identification, should modified to be:

Isolation and identification of bacterial pathogens

-Add specific references to the Isolation and identification of bacterial pathogens, enumerate the used biochemical reactions, add the company names and countries of the used bacterial media.

- Update the CLSI reference, use 2018 version.

Antimicrobial susceptibility testing. Besides, please illustrate in the text the reasons for the selection of these antimicrobial agents.

-Multilocus sequence typing (MLST) should be performed to illustrate the genetic relatedness between the recovered isolates. If it's not applicable please add it to the study limitations.

-Results:

- Good presentation; however, please support your results with illustrating figures.

-Table 2: -Please write the scientific names of bacterial pathogens in correct form (italic form).

-Discussion:

- The discussion is good; but the authors are advised to illustrate the real impact of their findings without repetition of results.

-Conclusion

- A real conclusion should focus on the question or claim you articulated in your study, whose resolution has been the main objective of your paper? That question now needs to be re-invoked and definitively answered. More still, you need to leave your reader with a higher level of insight into your topic

6. PLOS authors have the option to publish the peer review history of their article (what does this mean?). If published, this will include your full peer review and any attached files.

Reviewer #1: No

---

## [Author Response · Author response to Decision Letter 0]

2 Aug 2021

Response to 1: The manuscript is technically sound and the data support the conclusions.

Response to 2: The statistical analysis has been performed appropriately and rigorously.

Response to 3: The authors have made all data underlying the findings in their manuscript fully available.

Response to 4: The manuscript is presented in an intelligible fashion and written in standard English. We have edited our manuscript with the AJE which PLOS ONE recommended, thoroughly copyedit our manuscript for language usage, spelling, and grammar.

Response to 5: We have revised language editing and grammar mistakes in the manuscript. The scientific names of bacterial pathogens were writen in italic form.

1) Title: We have modify the title.

2) Abstract: We have revised the abstract to include the main conclusions of the study.

3) Introduction: We have added the virulence determinants, the pathogenesis, and diseases caused by S. aureus, E. coli, Acinetobacter baumannii, and K. pneumonia in instroduction. We added paragraphs given by reviewers to the manuscript and cited references. The purpose of the study was restated.

4) Material and methods: The data on the origin of the specimen, due to the loss of data 13 years ago, only data from 2018 to 2020, so it is not listed in the manuscript. The data of the last two years is shown in the following tables.

5) The subtitle: Bacteria isolation and identification have modified to be: Isolation and identification of bacterial pathogens. We have added the company names and countries of the used bacterial media. The biochemical reactions according to instructions. We have update the CLSI reference use 2018 version. Multilocus sequence typing it's not applicable in our study.

6) Results: The scientific names of bacterial pathogens were in italic form. 

7) Discussion: We have revised the discussion as requested. 

8) Conclusion: We have revised the discussion as requested.

Response to 6: No peer review of published articles.

See the document doc. response to reviewer.

---

## [Editor Report · Decision Letter 1]

5 Aug 2021

Comparative study on the changes of bacterial species and severity of antimicrobial resistance during 13 years

PONE-D-21-10409R1

Dear Dr. Zhang,

We’re pleased to inform you that your manuscript has been judged scientifically suitable for publication and will be formally accepted for publication once it meets all outstanding technical requirements.

Kind regards,

Abdelazeem Mohamed Algammal, Prof, Ph.D

Academic Editor

PLOS ONE
---

## [Editor Report · Acceptance letter]

16 Aug 2021

PONE-D-21-10409R1 

Comparative study on the changes of bacterial species and severity of antimicrobial resistance during 13 years 

Dear Dr. Zhang:

I'm pleased to inform you that your manuscript has been deemed suitable for publication in PLOS ONE. Congratulations! Your manuscript is now with our production department. 

Kind regards, 

on behalf of

Professor Abdelazeem Mohamed Algammal 

Academic Editor

PLOS ONE